# Artificial intelligence–Driven detection and decision support system for precision management of maize downy mildew

Jadesha G.[1]*, Anurag Dhole[2], Deepak D. [2]*

**1** Plant Pathologist, College of Agriculture, GKVK, University of Agricultural Sciences, Bangalore, Karnataka, India, **2** Department of Mechatronics, Manipal Institute of Technology, Manipal Academy of Higher Education, Manipal, Karnataka, India

\* jadesha.uasb@gmail.com (JG); deepak.d@manipal.edu (DD)

## Abstract

Artificial intelligence (AI) enables rapid and precise plant disease detection, offering transformative potential for crop protection. Maize downy mildew (MDM), a destructive disease, causes substantial yield losses, making early detection critical. In this study, we evaluated the performance of thirteen machine-learning (ML) and deep-learning (DL) algorithms for classifying healthy and infected maize leaves using a curated field dataset. Model performance was assessed using multiple metrics, including accuracy, precision, recall, F1-score, and AUC-ROC. Among the tested models, VGG16 achieved the highest performance, with 97% accuracy, 0.98 precision, 0.95 recall, 0.97 F1-score, and an AUC-ROC of 0.99. Training and validation curves indicated minimal overfitting, demonstrating robust generalization. Feature visualization using t-SNE revealed clear separability between healthy and diseased samples, while Grad-CAM analysis confirmed that VGG16 focused on biologically relevant symptomatic regions, such as chlorotic streaks and leaf discoloration. Confusion matrix analysis further validated near-perfect classification, with very few misclassifications. Furthermore, we developed a web-based application (https://maize-mdm.streamlit.app/) that not only classifies MDM but also provides farm-level advisory measures. Two-year field trials of DSS-guided fungicide applications effectively suppressed MDM, reducing disease severity (PDI 3.20–5.20; PROC 93−96%), increasing grain yield (75.6–80.2 q/ha; PIOC 195−289%), and improving economic returns (B:C ratio 3.36–3.57) compared to untreated controls. Overall, this study demonstrates that AI-driven models, integrated with web-based decision support, provide accurate, interpretable, and actionable solutions for precision management of maize diseases, contributing to improved yield, profitability, and sustainable agricultural practices.

**Data availability statement:** We confirm that all data required to replicate the results of this study are fully provided within the manuscript. The manuscript contains the complete minimal data set underlying all reported analyses, including the values supporting summary statistics, tables, figures, and model performance metrics. No additional raw data beyond those presented in the manuscript were used in this study; therefore, no separate Supporting Information files are required.

**Funding:** Our research endeavor was supported by the Office of the Director of Research (grant number 9331/6330, 2022-23) University of Agricultural Sciences, Bangalore, Karnataka, India. The funders had no role in study design, data collection and analysis, decision to publish, or preparation of the manuscript.

**Competing interests:** The authors declare that they have no known competing financial interests or personal relationships that could have appeared to influence the work reported in this paper.

## Introduction

Maize (*Zea mays* L.) is the most important cereal crop worldwide, valued for its adaptability, versatility, and diverse uses. Globally, it is the leading cereal crop by production volume, serving as both food and feed, particularly in sub-Saharan Africa and Latin America [1]. In India, maize is the important cereal after rice and wheat in terms of area and production, contributing significantly to the agricultural economy. It serves as a staple food, a key input for the poultry and livestock sector, and a raw material for starch, ethanol, and other products [2]. With demand rising across multiple sectors, sustaining maize productivity is essential for both global and national food security.

Among the biotic stresses threatening maize cultivation, downy mildew is one of the most destructive diseases, mainly in tropical and subtropical Asia. The disease is caused by several species of the genus *Peronosclerospora*, including *P. sorghi* and *P. heteropogoni*, which exhibit variability in host specificity and geographical distribution [3]. Characteristic symptoms include chlorotic streaks, leaf stunting, and yield reductions under conducive environmental conditions. In some parts of Asia, downy mildew is an endemic problem, while in others it is classified as a high-risk foreign pathogen threat [4]. Its epidemiological significance and potential for rapid spread highlight the urgent need for early and reliable detection systems to safeguard maize production [5].

Conventional detection methods, such as visual inspection and laboratory-based diagnostics, are constrained by several limitations. They are often subjective, dependent on expert knowledge, time-consuming, and difficult to scale for large monitoring areas [5–8]. Inconsistent field diagnosis can delay control measures, worsening disease spread and yield losses. While laboratory tests provide precision, they are costly and impractical for continuous surveillance in vast maize-growing regions [9]. These limitations underscore the need for innovative, efficient, and scalable approaches for real-time detection of maize downy mildew.

Recent advances in artificial intelligence (AI), particularly machine-learning (ML) and deep-learning (DL), have transformed plant disease detection by enabling high-throughput, image-based classification of plant health status. Traditional ML methods, including Support Vector Machines (SVMs), Random Forests (RF), and Gradient Boosting Machines (GBM), laid the groundwork by leveraging manually extracted features such as texture and colour [10]. However, these approaches are often constrained in handling complex datasets and require extensive preprocessing. DL models, especially Convolutional Neural Networks (CNNs), surpass traditional ML in accuracy and robustness by enabling automated feature extraction and superior performance in image-based disease recognition [11,12]. Furthermore, transfer learning approaches have enhanced the efficiency of DL by adapting pre-trained architectures to agricultural datasets, addressing challenges of limited labelled data and improving generalization across environments [13].

Among CNN architectures, VGG16 has emerged as a reliable and widely used model for plant disease detection. Its deep architecture of 16 weight layers with small receptive fields effectively captures intricate leaf symptom details critical for accurate

classification [14,15]. VGG16 has steadily achieved high accuracy across diverse field datasets while maintaining computational efficiency and interpretability. Its compatibility with transfer learning further enhances its versatility for detecting subtle disease symptoms under field conditions [16,17]. In addition, interpretability tools such as Gradient-weighted Class Activation Mapping (Grad-CAM) allow visualization of symptom regions emphasized by the model, increasing trust and practical utility in agricultural contexts [18].

Despite the advances of ML and DL in plant pathology, their deployment in practical crop protection remains limited. Farmers and field stakeholders need accessible, user-friendly platforms that not only identify disease incidence but also deliver actionable recommendations for management. Web-based diagnostic tools that integrate AI models provide an effective pathway to bridge research and practice by enabling real-time detection and advisory services [5,6]. Unlike existing studies that primarily focus on image-based disease classification, this work uniquely integrates deep-learning–based detection with explainable AI techniques and validates a web-enabled decision support system through multi-season field trials, thereby bridging the gap between AI-driven diagnostics and practical maize disease management.

In this context, the present study systematically evaluated the performance of thirteen ML and DL algorithms for classifying maize downy mildew using a curated image dataset. Relative analyses were made with multiple classification metrics, including accuracy, precision, recall, F1-score, and AUC-ROC. Among the tested models, VGG16 showed higher performance, achieving near-perfect classification with high stability across training and validation. Advanced interpretability analyses, including t-SNE visualization and Grad-CAM, confirmed its robustness in feature extraction and biologically relevant symptom recognition. To translate these research outputs into practice, the trained VGG16 model was deployed in a web application (https://maize-mdm.streamlit.app/), which not only classifies maize leaf health status but also delivers farm-level advisory for effective downy mildew management. Furthermore, the web-based advisory system was validated in field conditions, where fungicide applications guided by the DSS effectively reduced disease severity, improved yield advantage, and enhanced economic returns as reflected in higher benefit–cost (B:C) ratios. This integrated framework demonstrates the potential of AI-driven solutions and DSS-guided advisories to strengthen disease surveillance, management, and decision support in maize cultivation.

## Materials and methods

The methodological framework employed in this study is illustrated in Fig 1, depicting the systematic workflow for MDM diagnosis using ML and DL approaches. The methodology encompasses sequential steps of I. data acquisition and pre-processing, II. Model selection and Training, III. Performance evaluation, IV. Decision Support System. Each component is described in detail below.

### I. Dataset acquisition and pre-processing

**Image acquisition.** Maize leaf images representing both healthy and MDM-infected specimens were systematically collected under diverse field conditions and controlled environments to ensure dataset representativeness and robustness. Field photographs were captured from commercial maize farms across multiple agro-climatic zones, research plots, and greenhouse facilities where natural MDM outbreaks caused by *P. sorghi* were documented. The image dataset was collected across two consecutive cropping seasons (Kharif 2023 and Kharif 2024) from maize-growing locations in Southern Karnataka, covering Agro-climatic Zones V and VI. Images were acquired at different crop growth stages to capture temporal variability in disease expression under field conditions. Images were acquired using high-resolution digital cameras (minimum 12 MP) with standardized protocols to maintain consistent image quality and minimize background interference. The comprehensive dataset comprised images captured from our own field trails under varying lighting conditions and growth stages to enhance model generalizability. The final dataset included balanced representation of 3100 healthy and 3152 diseased samples to prevent class imbalance issues during training.

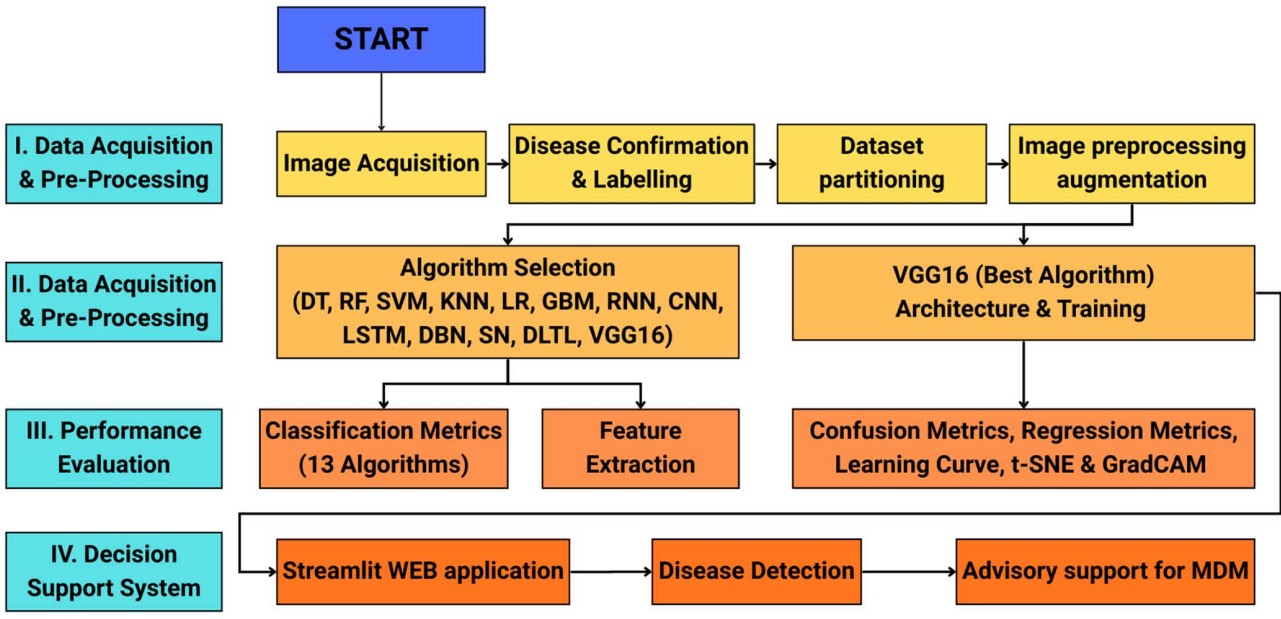

**Fig 1. Workflow framework for diagnosing MDM and providing field advisory.**

**Disease confirmation and labelling.** Plant pathologists and certified disease specialists manually annotated all images following established MDM diagnostic protocols. Quality control measures included double-blind validation by independent experts to ensure labelling accuracy. Images exhibiting ambiguous symptoms or poor quality were excluded from the dataset. The labelling process incorporated binary classification (healthy vs. diseased) to support comprehensive model training and evaluation.

**Dataset partitioning.** The complete dataset consisted of 6,252 maize leaf images, including 3,100 healthy and 3,152 MDM-infected samples. The dataset was partitioned into training, validation, and independent test sets using a 75:15:15 split. Accordingly, 4,689 images were used for model training, 938 images for validation, and 938 images for independent testing. The independent test set was not used during model training or hyperparameter tuning and was reserved exclusively for final performance evaluation.

**Image pre-processing and augmentation.** To improve model robustness and standardize the dataset, image pre-processing techniques were used. To retain optimal computational efficiency and assure compatibility with Deep-learning architectures, all images were scaled to 224 × 224 pixels. To provide reliable training convergence, min-max scaling was used to standardize pixel values to the range. Numerous data augmentation approaches, such as random rotation (0–360° in 30° increments), flipping horizontally and vertically, brightness correction (±20%), and contrast enhancement (0.8–1.2×), were used to expand the dataset size and strengthen the model's generalization capabilities.

## II. Model selection and training

**Algorithm selection and implementation.** Thirteen distinct machine-learning and deep-learning algorithms were systematically evaluated for MDM classification to identify optimal approaches. Traditional machine-learning methods included Decision Trees, Random Forest, Support Vector Machines, K-Nearest Neighbors, Logistic Regression, Gradient Boosting Machines, and XGBoost. Advanced neural network architectures comprised Artificial Neural Networks with multiple hidden layers, Convolutional Neural Networks, Long Short-Term Memory networks, Deep Belief Networks, and Siamese Networks along with Deep-learning approaches with transfer learning included VGG16 were used.

**VGG16 architecture implementation.** The VGG16 architecture was selected as the primary Deep-learning model based on its proven effectiveness in plant disease classification tasks. The pre-trained VGG16 model, originally trained on ImageNet dataset, was fine-tuned using transfer learning techniques specifically for MDM detection. The architecture consisted of 13 convolutional layers with 3×3 filters, 5 max-pooling layers with 2×2 windows, and 3 fully connected layers with dropout regularization (p = 0.5). Rectified Linear Unit (ReLU) activation functions were employed in hidden layers to introduce non-linearity, while the final output layer utilized softmax activation for probability distribution computation. Batch normalization was implemented between convolutional layers enhancing training speed and stability.

**Software environment and implementation.** The experimental framework was implemented using Python 3.8.10 with specialized libraries for ML and DL applications. TensorFlow 2.6.0 and Keras 2.6.0 provided the primary Deep-learning framework, while PyTorch 1.9.0 was utilized for specific model implementations. OpenCV 4.5.3 handled image processing operations, and Scikit-learn 1.0.2 supported traditional ML algorithms and evaluation metrics. NumPy 1.21.0 and Pandas 1.3.3 facilitated numerical computations and data manipulation. Matplotlib 3.4.3 and Seaborn 0.11.2 were employed for visualization and statistical plotting.

**Hyperparameter optimization.** To optimize model performance, multiple hyperparameter tuning strategies were employed. Grid search and random search were used to identify optimal hyperparameter combinations, while adaptive learning rate adjustments through the Adam optimizer ensured stable convergence. For training the VGG16 model, a batch size of 32 was used, and the model was trained for a maximum of 10 epochs. The initial learning rate was set to 0.0001. Early stopping based on validation loss was applied to prevent overfitting, and the model with the best validation performance was retained for final evaluation. Furthermore, batch normalization and dropout were incorporated to enhance generalization and reduce overfitting.

## III. Performance evaluation

**Comparison of classification metrics.** Model performance was measured using a combination of classification and regression-based metrics to ensure a robust and comprehensive assessment. For all thirteen algorithms, standard classification metrics were employed, including Accuracy, Precision, Recall, F1-Score, AUC-ROC, Specificity, and Negative Predictive Value (NPV). These metrics collectively gave insights into both overall performance and class-wise discrimination, accounting for the trade-off between sensitivity and specificity.

Accuracy reflects the proportion of correctly predicted disease occurrences relative to total predictions, thereby quantifying overall predictive correctness [19]. Precision, defined by Powers [20], measures the fraction of true positive predictions among all positive predictions, indicating the reliability of positive classifications and minimizing false positives. Recall, also described by Powers [20], quantifies the proportion of true positives identified among all actual positives, reflecting the model's capacity to capture disease instances. To balance these measures, the F1-Score was employed, representing the harmonic mean of precision and recall and providing a single robust indicator of classification performance [14]. In addition, AUC-ROC was used to evaluate discriminative power across varying thresholds, while Specificity (true negative rate) and NPV offered complementary insights into the models' ability to correctly identify and exclude healthy samples. A confusion matrix was also generated for each model, summarizing prediction outcomes against actual observations for a detailed performance breakdown [19].

**Comparison of feature extraction.** Feature extraction was performed to measure the capability of different ML and DL algorithms to capture discriminative patterns from maize leaf images affected by MDM. All input images were pre-processed through resizing and normalization to ensure consistency in scale and pixel distribution before being used for feature extraction. For DL models, particularly CNN, features were derived from the final convolutional or fully connected layers, while traditional ML algorithms utilized handcrafted feature representations generated from the pre-processed images.

To maintain fairness, the same dataset split was applied across all models, ensuring consistent evaluation of their feature extraction competences. This approach enabled a relative analysis of representational strength across ML and DL

methods, highlighting their efficacy in distinguishing healthy from diseased samples. Including feature extraction analysis in the methodology provided a benchmark for model selection and interpretability, confirming that the most robust and biologically meaningful features were leveraged for downstream classification tasks.

**Regression metrics analysis of VGG 16.** For the most effective deep-learning algorithm (VGG16), additional regression-based metrics were applied as supplementary indicators of prediction behaviour. These included Mean Absolute Error (MAE), Mean Squared Error (MSE), Root Mean Squared Error (RMSE), and the Coefficient of Determination ($R^2$), as described in earlier AI-enabled DSS studies [5,6].

The VGG16 model produces continuous probability scores (ranging from 0 to 1) for the *diseased* class at the output layer. These probability scores were used as the predicted values, while the corresponding ground-truth labels were encoded as 0 (Healthy) and 1 (Diseased) and used as the actual reference values. Regression-based metrics were computed between the predicted probabilities and actual labels to quantify error magnitude, variance explanation, and prediction bias.

These metrics were used only as complementary indicators of prediction stability and confidence calibration and were interpreted alongside standard classification metrics (accuracy, precision, recall, F1-score, and AUC–ROC), which remain the primary measures of classification performance.

**Learning curve analysis of VGG 16.** To evaluate how model performance varies with training set size, a learning-curve analysis was performed for the VGG16 classifier and the best-performing traditional machine-learning models. The training data were incrementally sampled in 10% steps from 10% to 100% of the total training set, while the validation set remained fixed. For each subset, the models were trained using the established training pipeline, and the corresponding training and validation performance values produced during each run were collected automatically by the framework. These values were plotted against the proportion of training data using Matplotlib 3.4.3 to generate the learning-curve graphs.

**t-SNE visualization of VGG 16.** The VGG16 model's trained high-dimensional feature representations were shown using t-distributed Stochastic Neighbor Embedding (t-SNE) analysis. To produce two-dimensional visuals that demonstrated the model's ability to differentiate between healthy and MDM-infected maize samples, deep features were taken from the network's last layer and processed using t-SNE. It was possible to determine if healthy and diseased samples formed separate, separable clusters in the feature space by using color-coded markers to denote different classes in the generated scatter plots. To verify the model's learning efficacy, cluster quality was assessed using common metrics. Well-separated clusters showed that VGG16 effectively recorded significant disease-specific patterns that could consistently distinguish between healthy and diseased maize leaves.

**Grad-CAM analysis of VGG 16.** Gradient-weighted Class Activation Mapping (Grad-CAM) was applied to provide interpretable visual explanations of the VGG16 model's decision-making process. This technique generated class-discriminative localization maps that highlighted the most important image regions influencing classification decisions by analysing gradient information from the final convolutional layer. The resulting heatmaps were overlaid onto original maize leaf images using color-coded intensity maps, where high-intensity regions (red/yellow) indicated areas strongly associated with MDM disease symptoms and low-intensity regions (blue/green) represented areas with minimal diagnostic relevance. The visualization process ensured that heatmaps matched original image dimensions for accurate interpretation, while quantitative analysis measured the percentage of highly activated pixels to assess how precisely the model focused on disease-affected areas. This approach provided biological validation by confirming that the model's attention was directed toward actual symptom locations rather than irrelevant background features, thereby increasing confidence in the automated diagnostic system's reliability for real-world applications.

## IV. Development of decision support system with VGG 16

A Decision Support System (DSS) was developed using the Streamlit framework, chosen for its lightweight deployment, real-time response, and user-friendly design (Fig 2). The web-based platform is intended for farmers, agricultural

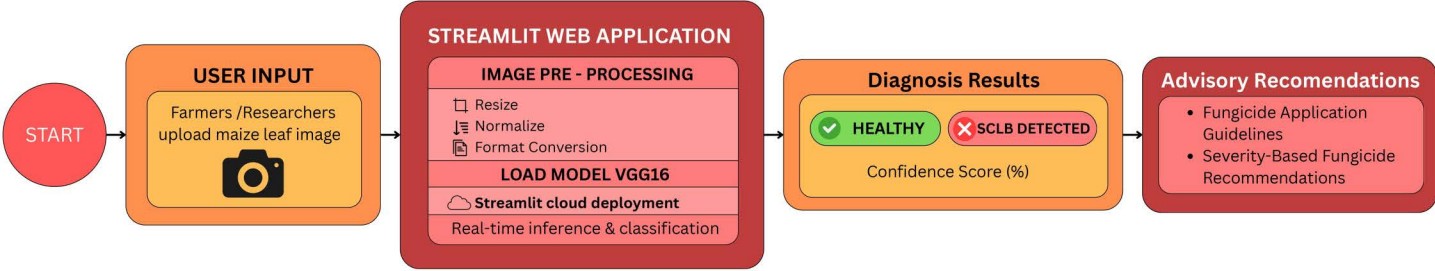

**Fig 2. Architecture of the DSS employing VGG16 for MDM identification.**

extension workers, and researchers, providing a simple interface for uploading maize leaf images to determine disease status. Once an image is uploaded, it undergoes a series of automated pre-processing steps such as resizing, normalization, and format conversion to ensure compatibility with the fine-tuned VGG16 model trained on MDM datasets. These steps help maintain consistency and accuracy across a wide range of input images. The processed images are then analysed by the VGG16 model, which performs real-time inference to classify each sample as either healthy or MDM-infected. To enhance transparency, the system also generates confidence scores that indicate the reliability of each prediction. The results are displayed through a clear, user-friendly interface. Beyond diagnosis, the DSS also provides practical advisory recommendations, including fungicide application guidelines and preventive agronomic practices, enabling timely and effective management of MDM in maize.

### V. Avoidable yield loss in maize through advisory measures for MDM management

To evaluate the effectiveness of Decision Support System (DSS)–guided advisory measures for managing maize downy mildew (MDM), field trials were conducted during the Kharif seasons of 2023 and 2024. The DSS integrates the trained VGG16 model to analyse field-acquired maize leaf images and confirm the presence of MDM infection. Fungicide application was initiated only after DSS-based disease confirmation, whereas untreated control plots did not receive any fungicide intervention.

Two treatments were included: (i) DSS-guided fungicide application comprising Metalaxyl 8% + Mancozeb 64% WP seed treatment @ 0.3% followed by Azoxystrobin 18.2% + Difenoconazole 11.4% SC foliar spray @ 0.1%, and (ii) an untreated control (UTC). Each treatment consisted of 12 replications, with each replication represented by a plot measuring 5 m in length with six rows, resulting in 12 plots per treatment.

The DSS provided advisory recommendations regarding the necessity and timing of fungicide application based on confirmed disease detection, but did not prescribe plot-wise spatial spraying within fields. Thus, the field trials evaluated the effectiveness of DSS-triggered advisory intervention rather than uniform or calendar-based fungicide application.

Observations on percent disease incidence (PDI) were recorded 45 days after sowing (DAS), and percent reduction over control (PROC) was calculated to quantify disease suppression. Grain yield (q/ha) was recorded at harvest to assess yield advantage and cost–benefit ratio. Statistical analyses using paired t-tests were performed to compare PDI, PROC, and grain yield between DSS-guided treated plots and untreated control plots, providing statistical validation of the effectiveness of DSS-based advisory measures for MDM management.

## Results

### Machine-learning and deep-learning models in classifying MDM disease

The classification performance of thirteen machine-learning and deep-learning algorithms was evaluated for the detection of maize downy mildew disease using multiple metrics, including accuracy, precision, recall, F1-score, and AUC-ROC (Table 1; Figs 3 and 4).

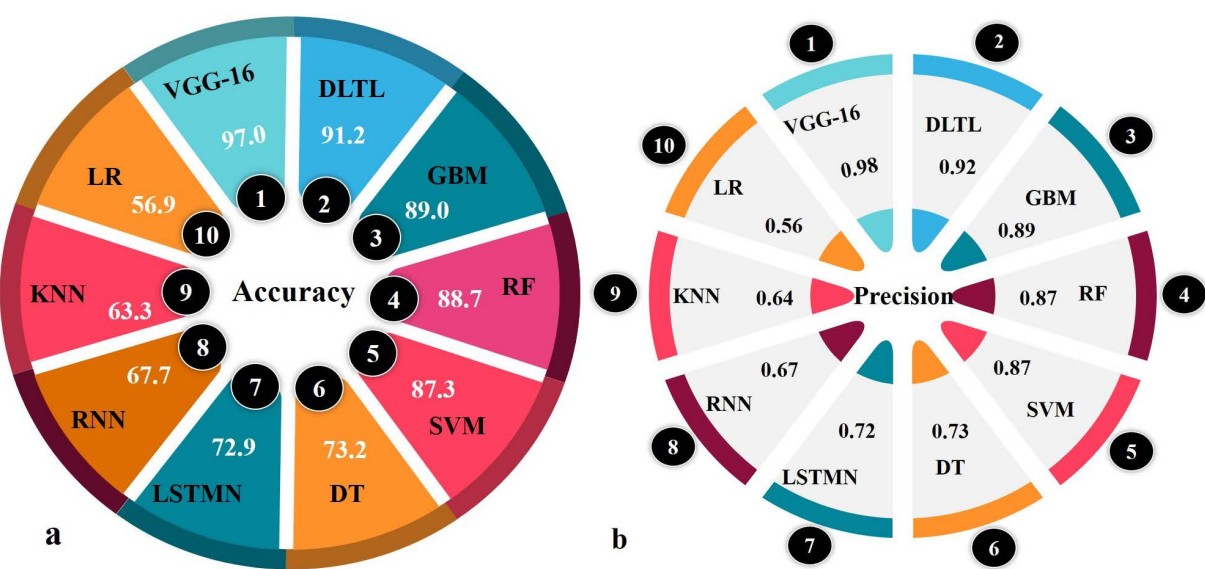

**Table 1. Comparative performance of thirteen machine-learning and deep-learning models for maize downy mildew detection.**

| S. No | ML Algorithm | Accuracy | Precision | Recall | F1 Score | AUC-ROC |
|---|---|---|---|---|---|---|
| 1 | Siamese Networks | 42.4 | 0.00 | 0.00 | 0.00 | 0.50 |
| 2 | Convolutional Neural Networks | 52.0 | 0.59 | 0.47 | 0.52 | 0.49 |
| 3 | Deep Belief Networks | 55.0 | 0.30 | 0.54 | 0.39 | 0.49 |
| 4 | Logistic Regression | 56.9 | 0.56 | 0.56 | 0.56 | 0.55 |
| 5 | K-Nearest Neighbors | 63.3 | 0.64 | 0.63 | 0.60 | 0.60 |
| 6 | Recurrent Neural Networks | 67.7 | 0.67 | 0.67 | 0.67 | 0.60 |
| 7 | Long Short-Term Memory Networks | 72.9 | 0.72 | 0.72 | 0.72 | 0.71 |
| 8 | Decision Trees | 73.2 | 0.73 | 0.73 | 0.73 | 0.72 |
| 9 | Support Vector Machines | 87.3 | 0.87 | 0.87 | 0.87 | 0.87 |
| 10 | Random Forest | 88.7 | 0.87 | 0.93 | 0.90 | 0.88 |
| 11 | Gradient Boosting Machines | 89.0 | 0.89 | 0.89 | 0.89 | 0.89 |
| 12 | Deep Learning with Transfer Learning | 91.2 | 0.92 | 0.91 | 0.91 | 0.91 |
| 13 | VGG16 | **97.0** | **0.98** | **0.95** | **0.97** | **0.99** |

All metrics were computed using the independent test set.

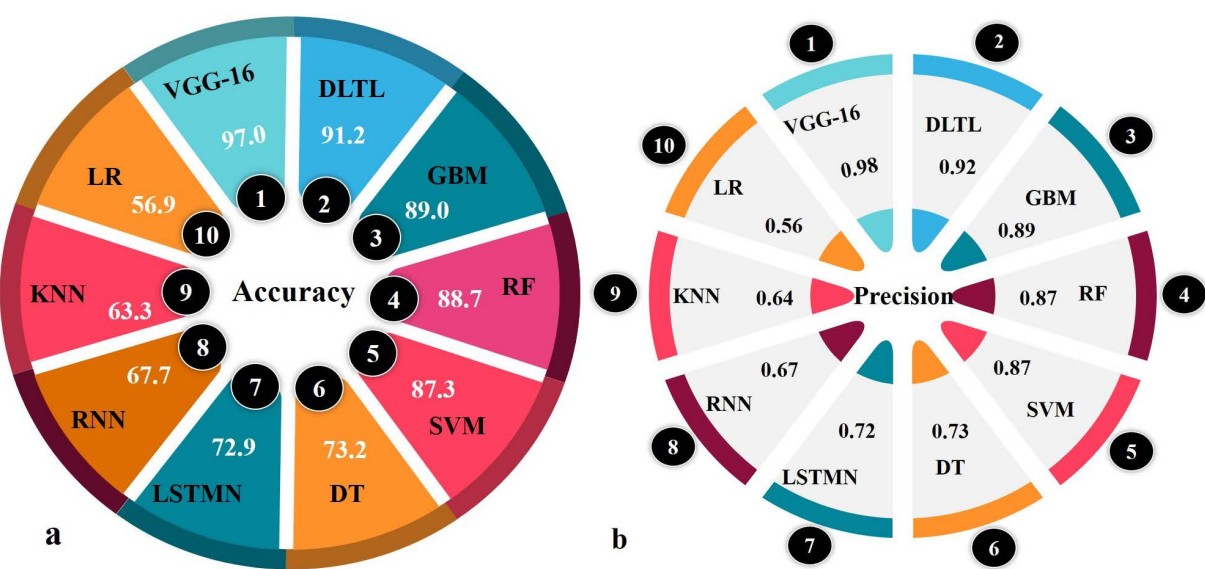

**Fig 3. Comparative accuracy and precision of the top 10 models for MDM recognition.**

Among the conventional ML models, Logistic Regression (56.9% accuracy; AUC-ROC = 0.55), K-Nearest Neighbors (63.3%; AUC-ROC = 0.60), and Decision Trees (73.2%; AUC-ROC = 0.72) showed moderate predictive capability. Support Vector Machines (87.3%; AUC-ROC = 0.87) markedly outperformed other traditional algorithms, reflecting its robustness in handling complex disease image classification. Ensemble learning methods further improved prediction performance. Gradient Boosting Machines (89.0% accuracy; AUC-ROC = 0.89) and Random Forest (88.7% accuracy; AUC-ROC = 0.88) consistently achieved strong results, with Random Forest showing the highest recall (0.93), suggesting better sensitivity in disease detection.

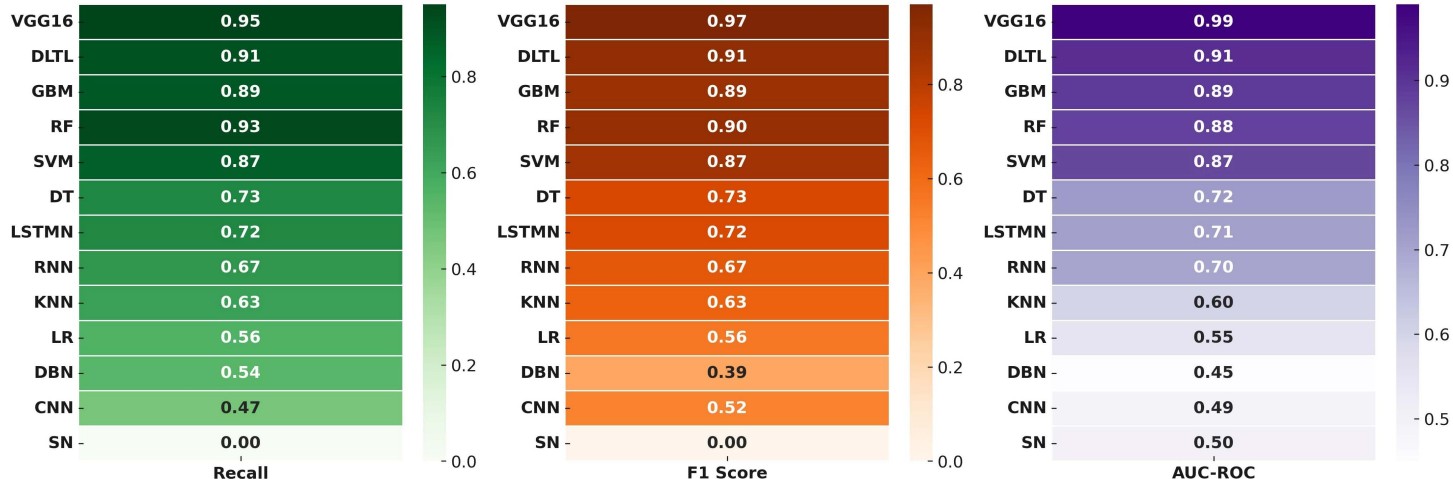

**Fig 4. Evaluation of recall, F1-score, and AUC–ROC metrics for MDM detection models.**

Deep-learning models generally performed better than classical ML algorithms. Recurrent Neural Networks (67.7%; AUC-ROC = 0.60) and Long Short-Term Memory Networks (72.9%; AUC-ROC = 0.71) showed moderate improvements in recall and F1-score, while Deep Belief Networks (55.0%; AUC-ROC = 0.49) and Convolutional Neural Networks without transfer learning (52.0%; AUC-ROC = 0.49) showed limited performance. Siamese Networks performed poorly with only 42.4% accuracy and no meaningful classification ability (precision, recall, and F1-score = 0.00).

Transfer learning-based approaches proved to be highly effective. Deep-learning with Transfer Learning achieved 91.2% accuracy with an AUC-ROC of 0.91, substantially outperforming most ML and DL models. The VGG16 architecture emerged as the best-performing model, achieving 97.0% accuracy, 0.98 precision, 0.95 recall, 0.97 F1-score, and an AUC-ROC of 0.99. These results indicate near-perfect classification capability, establishing VGG16 as the most reliable model for maize downy mildew detection. Overall, the findings highlight the superiority of deep-learning models with transfer learning, particularly VGG16, in delivering accurate and consistent classification of maize downy mildew disease images compared to conventional ML and shallow DL approaches.

## Feature extraction scores across algorithms

The feature extraction performance of different machine-learning and deep-learning algorithms for maize downy mildew detection is presented in Fig 5.

Among all tested models, VGG16 achieved the highest score (0.99), demonstrating its superior capability in extracting discriminative features relevant to disease symptoms. This was closely followed by Deep Transfer Learning (DLTL, 0.97), Gradient Boosting Machine (GBM, 0.97), and Random Forest (RF, 0.97), which also exhibited strong performance. Support Vector Machine (SVM) attained a moderately high score (0.95), while Decision Tree (DT, 0.74) and Long Short-Term Memory (LSTM, 0.73) performed comparatively lower. Conventional models such as Logistic Regression (0.55), Recurrent Neural Network (0.56), and Deep Belief Network (0.40) displayed weaker performance, while basic Convolutional Neural Networks (CNN, 0.39) and simple statistical approaches (SN, 0.20) yielded the lowest feature extraction scores. Overall, the results highlight the effectiveness of deep transfer learning approaches, particularly VGG16, in capturing disease-relevant features from maize leaf images, establishing their suitability for robust disease forecasting and detection.

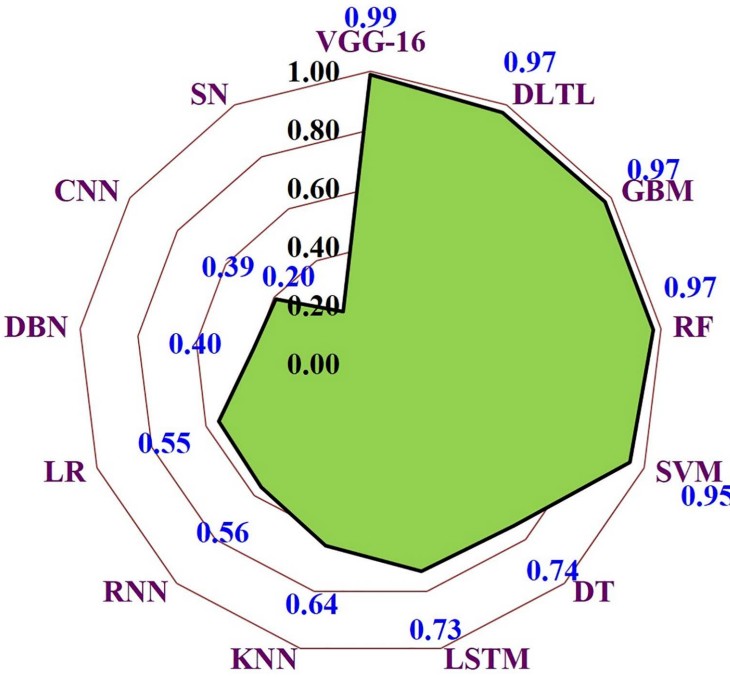

**Fig 5. Algorithmic comparison for feature extraction from MDM image datasets.**

## Evaluation of VGG16 classification accuracy using statistical indicators

The classification performance of the VGG16 model for MDM was further validated using multiple statistical indicators (Fig 6) was exclusively employed for this analysis, as it outperformed other models in the classification task. The model achieved an $R^2$ value of 0.93 and an explained variance score of 0.93, confirming strong agreement between the predicted and actual class outputs. Error-based measures also supported the reliability of the model. The Mean Absolute Error (MAE) was 0.08, the Mean Squared Error (MSE) was 0.03, and the Root Mean Squared Error (RMSE) was 0.17, reflecting minimal deviations in classification outcomes. Furthermore, the Mean Bias Deviation (MBD) was −0.03, indicating negligible systematic bias in the predictions. Together, these results strengthen the robustness of the VGG16 model in accurately classifying MDM disease, providing confidence in its suitability for practical disease identification and management.

## Confusion matrix evaluation for healthy vs. infected maize leaves

The classification performance of the VGG16 model for MDM was further examined using a confusion matrix (Fig 7). The confusion matrix was generated using a representative subset of the independent test set (156 healthy and 117 infected maize leaf images; total = 273) to clearly illustrate classification outcomes and misclassification patterns. Within this subset, the model correctly identified 154 out of 156 healthy samples and 111 out of 117 infected samples, resulting in only two false positives (healthy leaves misclassified as infected) and six false negatives (infected leaves misclassified as healthy). This near-perfect classification reflects the robustness of VGG16 in distinguishing between healthy and infected maize leaves. The very low misclassification rate highlights its reliability for practical disease detection applications. All classification performance metrics reported in this study (accuracy, precision, recall, F1-score, and AUC–ROC) were computed using the complete independent test set and are not limited to the subset shown in the confusion matrix.

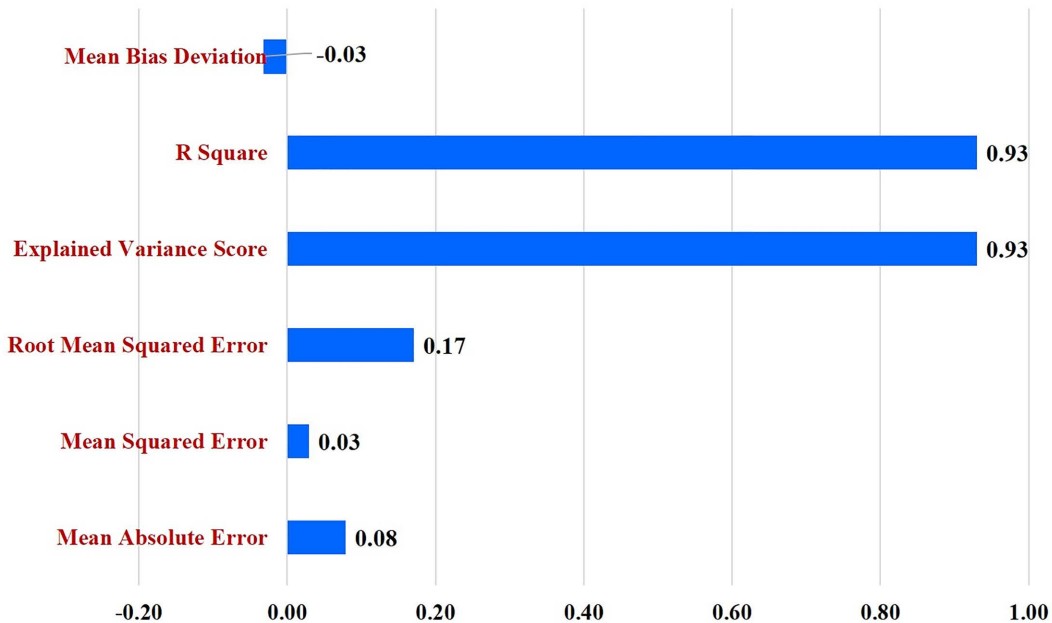

**Fig 6. Error rate and variance analysis of VGG16 model performance in MDM detection.**

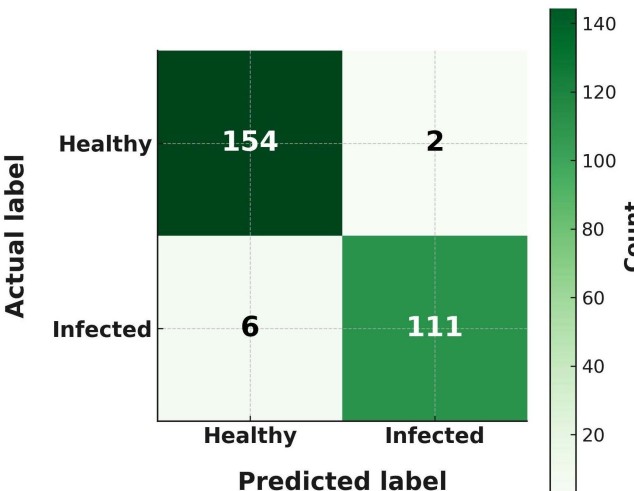

**Fig 7. Confusion matrix representation of VGG16 outcomes for MDM identification.**

### Training and validation accuracy of VGG16 across epochs

The learning behaviours of the best-performing model (VGG16) was evaluated across 10 epochs using training and validation accuracy (Fig 8). The training accuracy showed a sharp increase from 73% in the first epoch to above 90% by the third epoch, after which it continued to improve gradually, reaching approximately 96% at the tenth epoch. Validation accuracy also remained consistently high, starting at 90% in the first epoch and stabilizing around 93–95% over subsequent epochs. The close alignment between training and validation curves indicates minimal overfitting and demonstrates that the model generalized well to unseen data.

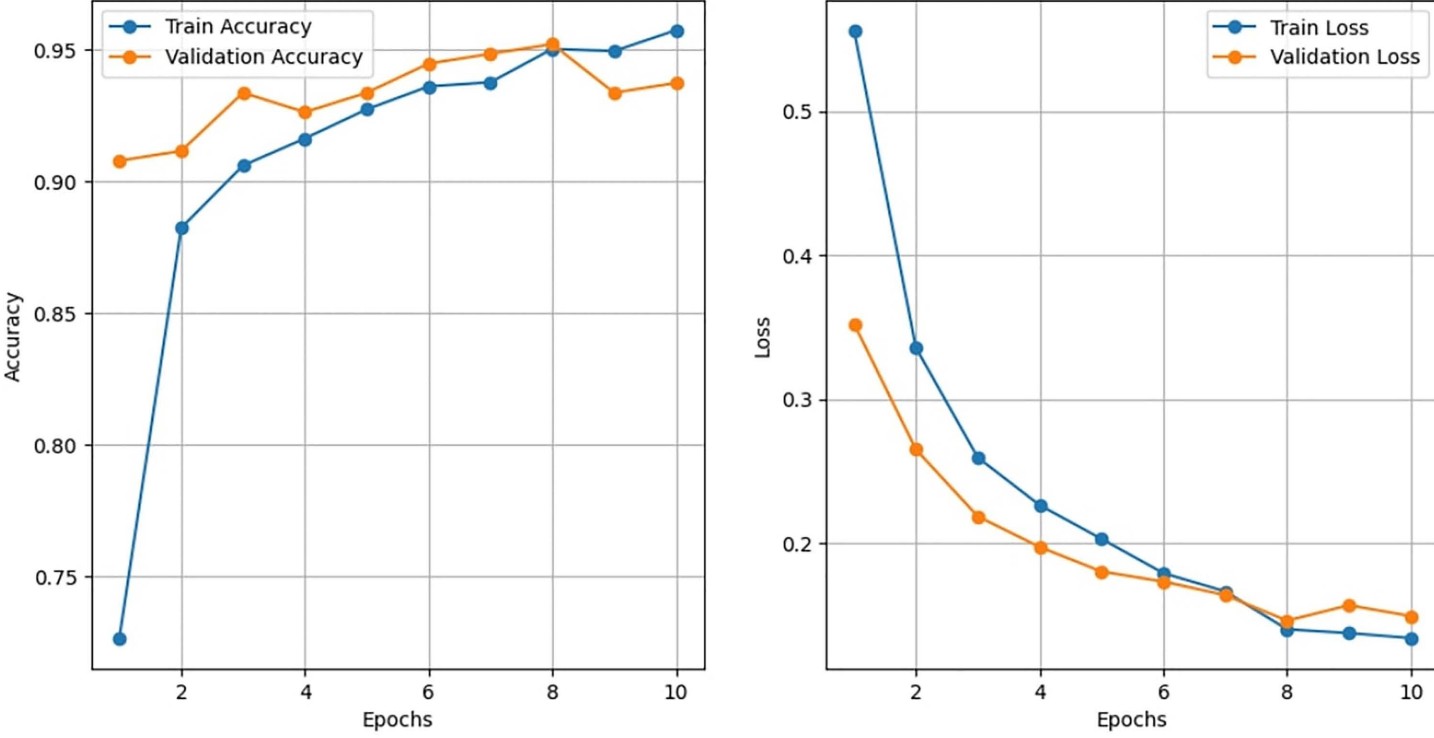

**Fig 8. Learning curve trends of VGG16 during MDM detection.**

These results confirm the stability and reliability of the VGG16 architecture in classifying maize downy mildew images, reinforcing its superior predictive performance compared to other machine-learning and deep-learning models.

### t-SNE visualization of learned features for maize downy mildew classification

To examine the discriminative power of the VGG16-extracted features, a t-distributed Stochastic Neighbour Embedding (t-SNE) plot was generated for maize downy mildew datasets (Fig 9). The visualization revealed two well-separated clusters corresponding to healthy and diseased samples. Healthy leaf images (blue) predominantly occupied the upper region, whereas infected leaf images (orange) clustered in the lower region with minimal overlap. This distinct separation highlights that the model effectively captured disease-specific representations, facilitating reliable classification of maize downy mildew. The clear boundaries observed in the t-SNE space further confirm the robustness of the deep-learning model in learning meaningful and non-redundant features for disease detection.

### Visualization of discriminative regions in MDM classification

To gain insights into the decision-making process of the VGG16 model for MDM detection, we employed Gradient-weighted Class Activation Mapping (Grad-CAM). Fig 10 illustrates the original diseased maize leaf image, the corresponding Grad-CAM heatmap, and the overlay of the heatmap on the original image. The Grad-CAM heatmap highlights the discriminative regions that contributed most strongly to the model's prediction. It can be observed that the network primarily focused on the symptomatic leaf areas exhibiting chlorotic streaks and yellowish-green discoloration, which are characteristic of maize downy mildew infection. The regions of high activation (represented by red areas in the heatmap) correspond to the disease-affected parts of the leaf, while the blue regions indicate areas of lower importance. The overlay image clearly demonstrates that

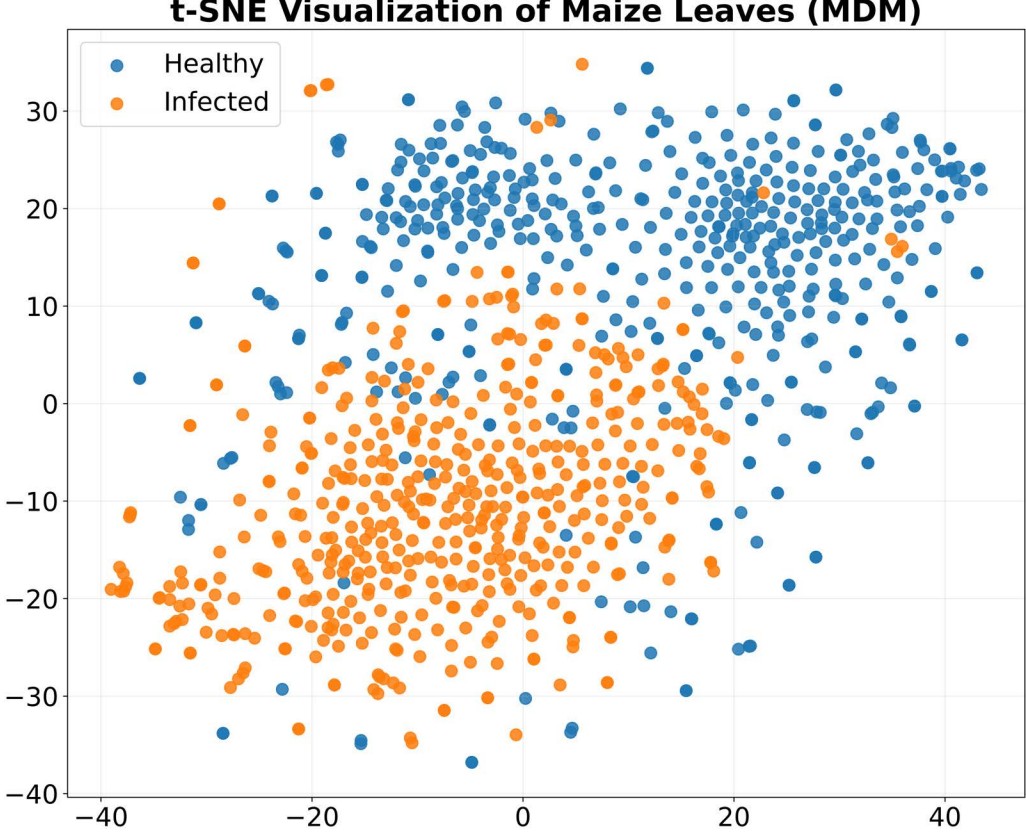

**Fig 9. t-SNE visualization of extracted features for MDM classification.**

VGG16 effectively localized disease-specific regions, confirming that the model is not only classifying the images accurately but also attending to biologically relevant features of maize downy mildew symptoms. This interpretability analysis further strengthens confidence in the reliability of the model for practical disease detection applications.

### Effectiveness of DSS-guided fungicide application in managing MDM and improving yield and economic returns

The management of maize downy mildew using Decision Support System (DSS)–guided advisory recommendations resulted in significant reductions in disease incidence (Table 2) and improvements in grain yield (Fig 11a) and economic returns (Fig 11b) compared to the untreated control. In this study, the DSS was used as a diagnostic and advisory trigger, whereby fungicide application was undertaken only after DSS-confirmed disease detection, while untreated control plots received no fungicide intervention.

In 2023, plots receiving DSS-guided fungicide intervention comprising Metalaxyl 8% + Mancozeb 64% WP as seed treatment (0.3%) followed by Azoxystrobin 18.2% + Difenoconazole 11.4% SC foliar spray (0.1%) recorded a percent disease incidence (PDI) of 3.20, corresponding to a percent reduction over control (PROC) of 95.6%. In contrast, untreated control plots recorded a PDI of 72.50. DSS-guided intervention resulted in a grain yield of 75.60 q/ha, representing a percent increase in yield over control (PIOC) of 195.3%, with a benefit–cost (B:C) ratio of 3.36 compared to 1.28 in untreated plots.

During 2024, the same DSS-guided advisory-based intervention resulted in a PDI of 5.20 and PROC of 93.6%, with a grain yield of 80.20 q/ha, corresponding to a PIOC of 289.3% over the control. The B:C ratio for DSS-guided treated plots was 3.57, whereas untreated plots recorded a B:C ratio of 1.03.

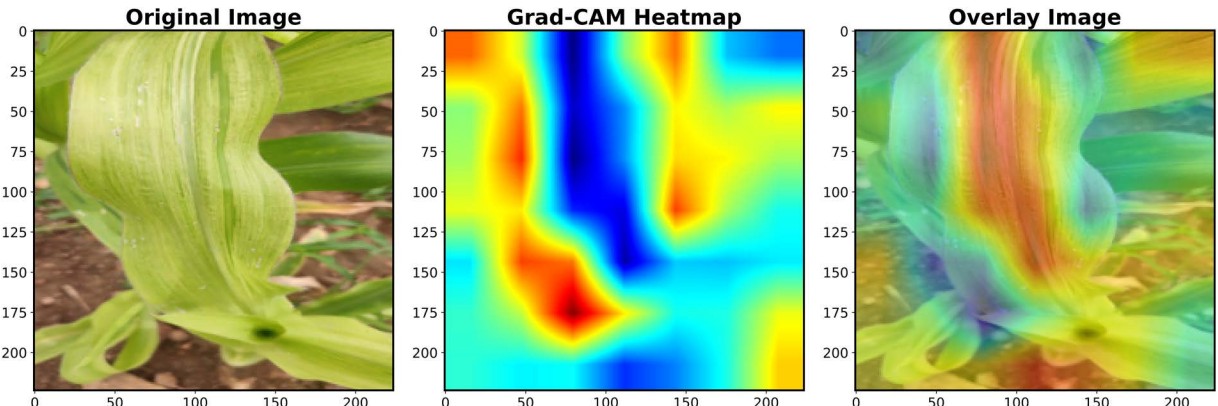

**Fig 10. Grad-CAM–driven interpretability analysis of VGG16 in MDM detection.**

**Table 2. Reducing Avoidable Yield Loss in Maize through Fungicide-Based Advisory Measures Against MDM Disease.**

| Year | Fungicide | PDI | PROC* (%) | Yield (q/ha) | PIOC# (%) | CB## Ratio |
|------|-----------|-----|-----------|--------------|-----------|------------|
| 2023 | Metalaxyl 8%+Mancozeb 64% WP Seed Treatment @ 0.3%+Azoxystrobin 18.2%+Difenoconazole 11.4% SC Foliar Spray @ 0.1% | 3.20 | 95.60 | 75.60 | 195.30 | 3.36 |
| | Untreated Control | 72.50 | | 25.60 | -- | 1.28 |
| 2024 | Metalaxyl 8%+Mancozeb 64% WP Seed Treatment @ 0.3%+Azoxystrobin 18.2%+Difenoconazole 11.4% SC Foliar Spray @ 0.1% | 5.20 | 93.60 | 80.20 | 289.30 | 3.57 |
| | Untreated Control | 81.20 | | 20.60 | -- | 1.03 |
| Mean | Metalaxyl 8%+Mancozeb 64% WP Seed Treatment @ 0.3%+Azoxystrobin 18.2%+Difenoconazole 11.4% SC Foliar Spray @ 0.1% | 4.20 | 94.50 | 77.90 | 237.2 | 3.46 |
| | Untreated Control | 76.85 | | 23.10 | -- | 1.16 |

\* PROC-Percent Reduction Over Control, #PROC-Percent Reduction Over Control, ##CB-Cost Benefit

Mean values across both years further confirmed the effectiveness of DSS-guided advisory-based fungicide management, with treated plots showing a PDI of 4.20, PROC of 94.5%, grain yield of 77.90 q/ha, PIOC of 237.2%, and a B:C ratio of 3.46, compared to untreated control plots with a PDI of 76.85, grain yield of 23.10 q/ha, and a B:C ratio of 1.16. These results demonstrate that DSS-triggered advisory intervention, rather than calendar-based or indiscriminate fungicide application, effectively suppresses maize downy mildew, enhances yield, and improves economic returns.

## Discussion

Maize (*Zea mays* L.) is a globally important cereal crop with major contributions to food security, livestock feed, and agro-industrial uses [5,6,21]. However, its productivity is severely constrained by downy mildew, caused by *Peronosclerospora* spp., particularly in tropical and subtropical regions [22,23]. Yield losses are often substantial, underscoring the need for reliable diagnostic and management strategies.

Conventional diagnostic methods, including field-based scoring and laboratory confirmation, are limited by subjectivity, cost, and scalability [24,25]. In recent years, machine-learning and deep-learning approaches have emerged as powerful alternatives for plant disease detection, providing rapid, accurate, and scalable solutions [11,26]. Our study compared thirteen ML and DL algorithms for classifying maize downy mildew and demonstrated the superiority of DL, particularly transfer learning models, over conventional ML approaches.

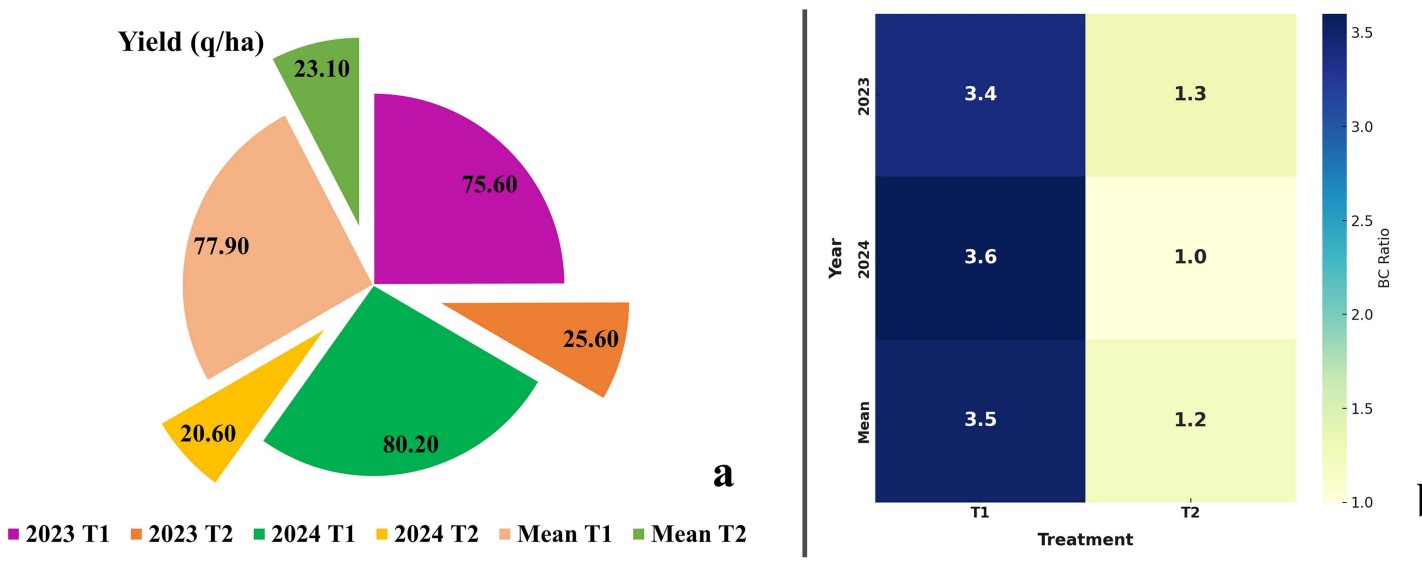

**Fig 11. Yield Advantage and Benefit-Cost Ratio Under Fungicide Advisory for MDM Management.**

Among ML models, Support Vector Machines achieved the highest performance (accuracy 87.3%, AUC-ROC 0.87), outperforming Logistic Regression, Decision Trees, and KNN. Ensemble algorithms such as Random Forest and Gradient Boosting Machines further improved classification, consistent with earlier reports that ensemble strategies reduce variance and enhance predictive reliability [9,27]. Despite these improvements, DL-based transfer learning consistently achieved higher accuracy and robustness.

VGG16 was the best-performing model, with an accuracy of 97.0%, precision of 0.98, recall of 0.95, F1-score of 0.97, and an AUC-ROC of 0.99. These results align with previous findings demonstrating the strong feature extraction capability of VGG16 for plant disease recognition [16,20]. Confusion matrix analysis confirmed near-perfect classification with minimal misclassifications. Grad-CAM visualization highlighted symptomatic chlorotic streaks, validating that the model focused on biologically relevant regions, thereby improving interpretability [18]. Learning curves indicated stable convergence with minimal overfitting, while t-SNE visualizations revealed distinct clustering of healthy and diseased samples, further confirming the discriminative power of the model [14,28].

Beyond model performance, integration of a decision support system (DSS) for fungicide advisory provided significant management benefits under field conditions. The DSS functioned as a diagnostic and advisory trigger, whereby fungicide application was undertaken following confirmed disease detection rather than calendar-based or indiscriminate application. The combination of Metalaxyl + Mancozeb seed treatment with Azoxystrobin + Difenoconazole foliar application resulted in a 94.5% reduction in disease severity, a 237.2% increase in grain yield, and an improved benefit–cost ratio (3.46 vs. 1.16 in untreated plots). These findings corroborate earlier reports on the effectiveness of triazoles and strobilurins in controlling downy mildew pathogens [12,19,29]. Overall, the results demonstrate that DSS-guided advisory-based intervention, when coupled with accurate AI-driven disease detection, can substantially enhance disease management outcomes and farm profitability.

Taken together, this study makes three key contributions. First, it establishes the superior accuracy and interpretability of DL-based transfer learning, particularly VGG16, for MDM detection. Second, it demonstrates the utility of visualization tools such as Grad-CAM and t-SNE in validating model predictions, thereby enhancing trust in AI-based diagnostics. Third, it shows that coupling AI-driven disease detection with DSS-guided fungicide management substantially improves

yield and economic returns. Collectively, these advances provide a scalable, sustainable framework for maize disease management and contribute toward improving food security.

### Impact, limitations, and future perspectives

The impact of this work lies in its integration of cutting-edge DL models with practical crop management. By achieving near-perfect classification of MDM symptoms and linking predictions to actionable fungicide advisories, the study bridges the gap between AI research and field-level applicability. The demonstrated yield gains and improved benefit-cost ratio highlight the potential of such integrated systems to reduce crop losses and enhance farmer livelihoods in maize-growing regions.

However, some limitations remain. First, while VGG16 proved effective, its computational requirements may limit real-time deployment on resource-constrained devices without optimization. Second, the DSS was evaluated under controlled field trials, and its scalability across larger geographies and diverse agro-climatic zones requires further validation.

Future research should address these gaps by (i) expanding training datasets with multi-location, multi-season image collections, (ii) exploring lightweight DL architectures and edge-AI solutions for mobile-based field diagnostics, (iii) integrating multimodal data such as weather forecasts and hyperspectral imaging to improve early detection, and (iv) validating DSS frameworks under participatory on-farm trials to ensure adaptability and farmer acceptance. Such advancements will strengthen the transition of AI-driven disease detection and management from research settings to scalable, real-world agricultural systems.

## Conclusion

This study demonstrates the potential of deep-learning–based approaches, particularly VGG16, for accurate and interpretable detection of maize downy mildew, achieving superior performance over conventional ML models. Integration of AI-driven disease classification with decision support system–guided fungicide advisories resulted in substantial reductions in disease severity, significant yield gains, and enhanced economic returns. By combining advanced diagnostic tools with practical management strategies, the work provides a scalable framework for sustainable maize health management. While further validation under diverse agro-climatic conditions and development of lightweight models for field deployment are necessary, the findings highlight a transformative pathway toward reducing crop losses, improving farmer profitability, and strengthening food security.

### Author contributions

**Conceptualization:** Jadesha G.

**Data curation:** Jadesha G.

**Formal analysis:** Jadesha G.

**Investigation:** Jadesha G.

**Methodology:** Jadesha G, Anurag Dhole.

**Resources:** Deepak D.

**Software:** Anurag Dhole.

**Supervision:** Deepak D.

**Visualization:** Anurag Dhole.

**Writing – original draft:** Jadesha G.

**Writing – review & editing:** Deepak D.

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
