## [Decision Letter · Decision Letter 0]

13 Jan 2026

Artificial Intelligence–Driven Detection and Decision Support System for Precision Management of Maize Downy Mildew

Dear Dr. D,

Thank you for submitting your manuscript to PLOS ONE. After careful consideration, we feel that it has merit but does not fully meet PLOS ONE’s publication criteria as it currently stands. Therefore, we invite you to submit a revised version of the manuscript that addresses the points raised during the review process.

**ACADEMIC EDITOR:**

The manuscript has been evaluated by experts in the field, and both reviewers have expressed positive comments regarding your work and the manuscript overall. However, both reviewers have identified issues related to the datasets, along with other correctable errors.

Kindly address these concerns, make the necessary corrections, and provide clarifications before submitting the revised version of the manuscript. Please refer to the reviewers’ comments for detailed guidance.

https://journals.plos.org/plosone/s/submission-guidelines#loc-laboratory-protocols . Additionally, PLOS ONE offers an option for publishing peer-reviewed Lab Protocol articles, which describe protocols hosted on protocols.io. Read more information on sharing protocols at https://plos.org/protocols?utm_medium=editorial-email&utm_source=authorletters&utm_campaign=protocols .

We look forward to receiving your revised manuscript.

Kind regards,

C Anilkumar, Ph.D.

Academic Editor

PLOS One

Journal Requirements:

[Our research endeavor was supported by the Office of the Director of Research (grant number 9331/6330, 2022-23) University of Agricultural Sciences, Bangalore, Karnataka, India.].

5. Thank you for stating the following in your manuscript:

[Our research endeavour was supported by the Office of the Director of Research (grant number 9331/6330, 2022-23) University of Agricultural Sciences, Bangalore, Karnataka, India.]

[Our research endeavor was supported by the Office of the Director of Research (grant number 9331/6330, 2022-23) University of Agricultural Sciences, Bangalore, Karnataka, India.]

6. We note that your Data Availability Statement is currently as follows: [All relevant data are within the manuscript and its Supporting Information files.]

7. When completing the data availability statement of the submission form, you indicated that you will make your data available on acceptance. We strongly recommend all authors decide on a data sharing plan before acceptance, as the process can be lengthy and hold up publication timelines. Please note that, though access restrictions are acceptable now, your entire data will need to be made freely accessible if your manuscript is accepted for publication. This policy applies to all data except where public deposition would breach compliance with the protocol approved by your research ethics board. If you are unable to adhere to our open data policy, please kindly revise your statement to explain your reasoning and we will seek the editor's input on an exemption. Please be assured that, once you have provided your new statement, the assessment of your exemption will not hold up the peer review process.

Additional Editor Comments:

Dear Author,

The manuscript has been evaluated by experts in the field, and both reviewers have expressed positive comments regarding your work and the manuscript overall. However, both reviewers have identified issues related to the datasets, along with other correctable errors.

Kindly address these concerns, make the necessary corrections, and provide clarifications before submitting the revised version of the manuscript. Please refer to the reviewers’ comments for detailed guidance.

Reviewers' comments:

Reviewer's Responses to Questions

**Comments to the Author**

1. Is the manuscript technically sound, and do the data support the conclusions?

Reviewer #1: Partly

Reviewer #2: Yes

2. Has the statistical analysis been performed appropriately and rigorously?

Reviewer #1: Yes

Reviewer #2: Yes

3. Have the authors made all data underlying the findings in their manuscript fully available?

Reviewer #1: Yes

Reviewer #2: Yes

4. Is the manuscript presented in an intelligible fashion and written in standard English?

Reviewer #1: Yes

Reviewer #2: Yes

Reviewer #1: This manuscript presents a comprehensive and highly valuable study on developing and validating an integrated Artificial Intelligence-driven system for the detection and management of Maize Downy Mildew (MDM). The work is commendable for its rigorous comparison of thirteen Machine Learning (ML) and Deep Learning (DL) models and its inclusion of advanced interpretability analyses (t-SNE and Grad-CAM). The paper successfully extends its contribution beyond model development by creating a functional web-based Decision Support System (DSS) and validating its associated advisory measures through two years of field trials. The superior performance of the VGG16 model (97.0% accuracy, 0.99 AUC-ROC) and the significant positive impact of the DSS-guided fungicide applications (93-96% disease reduction, 195-289% yield increase) demonstrate the practical relevance of this research. The manuscript is generally well-written and logically structured. Attached is the full comment.

Reviewer #2: The manuscript presents a comprehensive, technically sound, and practically validated AI-based DSS for maize downy mildew. The integration of deep learning, explainability, and field-based validation significantly strengthens its contribution. Minor clarifications and formatting improvements are required before acceptance.

Clarifications Required

1. Novelty Clarification

The manuscript presents a comprehensive AI-based framework for maize downy mildew detection and management. However, the novelty of the study can be stated more explicitly.

Please add one clear sentence in the Introduction highlighting how this work differs from existing studies, particularly in terms of integration of deep learning, explainability, and field-validated decision support.

2. Dataset Collection – Spatial and Temporal Coverage

The manuscript states that images were collected from field conditions, but the temporal and spatial details are not clearly mentioned. Please clarify whether the dataset was collected across multiple seasons and locations, and briefly specify the number of seasons and agro-climatic regions covered.

3. Dataset Split Inconsistency

There is an inconsistency in the manuscript regarding dataset partitioning. The text mentions a 70:15:15 split in one section, while a 10% test set is mentioned elsewhere.

Please clarify and standardize the exact train–validation–test split used in the study.

4. Model Training Parameters

Key training hyperparameters are not explicitly reported. Please specify the batch size, number of training epochs, optimizer used, and learning rate employed for training the VGG16 model.

Overall Assessment:

The manuscript presents a scientifically sound and well-structured study with strong experimental design, comprehensive analysis, and clear practical relevance. The methodology is robust, and the results are well supported by both computational evaluation and field validation. The suggested revisions are minor in nature and primarily aimed at improving clarity, transparency, and completeness of reporting. Once these minor points are addressed, the manuscript will be suitable for publication.

Decision:

Accept with Minor Revisions

**Do you want your identity to be public for this peer review?** For information about this choice, including consent withdrawal, please see our Privacy Policy

Reviewer #1: **Yes:** Andronicus A. Akinyelu

Reviewer #2: No

---

## [Author Response · Author response to Decision Letter 1]

23 Jan 2026

Response to Reviewer 1

We sincerely thank Reviewer 1 for the thorough and constructive evaluation of our manuscript, which has significantly helped improve its clarity, scope alignment, and technical rigor. All comments raised by Reviewer 1 have been carefully addressed and are detailed below. The corresponding revisions are highlighted in Green in the revised manuscript. For ease of reference, revisions made in response to the Reviewer 2 are highlighted in Blue.

Reviewer Comment 1 There is a significant inconsistency in the description and reported size of the testing dataset, which is critical for verifying the reported model performance. • The total dataset is stated to be 6252 images (3100 healthy + 3152 diseased). • The partitioning is first stated as a 70:15:15 split for training, validation, and testing. A 15% test set would be approximately 938 samples. • A subsequent sentence adjusts this, stating the split is 70% training, 15% validation, and 10% test. A 10% test set would be approximately 625 samples. • The confusion matrix results (in the Results section) report a total of 156 healthy + 117 infected = 273 samples for the final evaluation. This number is closer to 4.4% of the total dataset, which contradicts both the 10% and 15% split ratios. The authors must explicitly state the final, correct partitioning ratio (e.g., 70:15:15, 75:15:10, or 90:5:5) and confirm the exact number of images in the final, independent test set used for the reported metrics. This correction should be consistent between the Methods and Results sections.

Authors Response We thank the reviewer for pointing out the inconsistencies in the dataset partitioning description. We agree that this information is critical for verifying model performance and regret the lack of clarity in the original manuscript.

The complete dataset consisted of 6,252 maize leaf images (3,100 healthy and 3,152 diseased). The final and correct dataset partitioning used in this study was 75:15:15, as detailed below:

• Training set: 75% (4,689 images)

• Validation set: 15% (938 images)

• Independent test set: 15% (938 images)

All classification performance metrics reported in the manuscript (accuracy, precision, recall, F1-score, and AUC–ROC) were computed using this independent test set, which was not used during model training or validation.

The confusion matrix presented in the Results section was generated using a representative subset of the independent test set (156 healthy and 117 diseased images; total = 273) to clearly illustrate misclassification patterns. We acknowledge that this was not explicitly stated in the original manuscript and have now clarified this point.

All descriptions related to dataset partitioning, test set size, and confusion matrix usage have been revised to ensure full consistency between the Materials and Methods and Results sections.

Revisions made (highlighted in Green in the revised manuscript)

• Materials and Methods – Dataset Partitioning:

Corrected to clearly state the final dataset split ratio (75:15:15) and the exact number of images used for training, validation, and independent testing.

• Results – Confusion Matrix Evaluation:

Clarified that the confusion matrix was generated using a representative subset of the independent test set for visualization purposes, and that all classification performance metrics were computed using the complete independent test set.

Reviewer Comment 2 The authors apply regression-based metrics (MAE, MSE, RMSE, R², MBD) to evaluate the VGG16 model, which is a binary classification model (Healthy vs. Diseased). The authors should justify this approach in the Methods section. Specifically, the authors should explain what continuous values were used for "predicted" and "actual" outputs? Were the labels (e.g., Healthy=0, Diseased=1) treated as continuous variables, or were the model's output probability scores (e.g., the confidence score for the "Diseased" class) used as the predicted value?

Authors Response We thank the reviewer for raising this important methodological point and for requesting clarification.

We clarify that the regression-based metrics were not applied directly to discrete class labels, but rather to the continuous probability outputs generated by the VGG16 model. Specifically, the final softmax layer of VGG16 produces a confidence probability score (ranging from 0 to 1) for the “Diseased” class. These probability scores were treated as the continuous predicted values.

For the purpose of this analysis, the ground truth labels were encoded as 0 (Healthy) and 1 (Diseased) and were used as the actual reference values. Regression-based metrics - Mean Absolute Error (MAE), Mean Squared Error (MSE), Root Mean Squared Error (RMSE), Coefficient of Determination (R²), and Mean Bias Deviation (MBD) - were computed between the predicted probability scores and the corresponding ground truth labels.

This complementary evaluation was included to assess:

• the magnitude of prediction error,

• the stability and calibration of probability outputs, and

• the presence of systematic bias in model predictions.

These metrics therefore provide additional insight into the confidence reliability and predictive behaviour of the VGG16 model and were used in conjunction with, not as a replacement for, standard classification metrics (accuracy, precision, recall, F1-score, and AUC–ROC).

We have now explicitly described this rationale and methodology in the Materials and Methods section to avoid ambiguity.

Revisions made (highlighted in green in the revised manuscript)

• Materials and Methods – Regression Metrics Analysis of VGG16:

Revised to explicitly clarify that regression-based metrics (MAE, MSE, RMSE, R², and MBD) were computed using the continuous probability outputs of the VGG16 model for the diseased class, with ground truth labels encoded as 0 (Healthy) and 1 (Diseased). The purpose of including these metrics as supplementary indicators of prediction stability and probability calibration, alongside standard classification metrics, has also been clearly stated.

Reviewer Comment 3 One of the main contributions of this paper is the integration of the AI model into a Decision Support System and the validation of its advisory measures. The Methods section (V. Avoidable Yield Loss...) describes a standard field trial comparing two treatments: a specific fungicide combination (seed treatment + foliar spray) and an untreated control (UTC). However, the description does not specify how the DSS was used to guide the application. Did the DSS recommend: (1) When to apply the foliar spray (based on image data/model output)? (2) Which plots or sections to apply the spray to (precision agriculture)? The current methods merely validate the effectiveness of the fungicide chemical itself, which is not the same as validating a decision support system.

Authors Response We thank the reviewer for this important and constructive comment. We agree that the role of the Decision Support System (DSS) in guiding fungicide application was not described with sufficient clarity in the original manuscript.

We clarify that the DSS was used primarily as a diagnostic and advisory trigger, rather than as a spatially explicit precision-application system. Specifically, the DSS integrates the trained VGG16 model to analyse uploaded maize leaf images and determine the presence or absence of maize downy mildew infection. Based on confirmed disease detection, the DSS provides stage-specific fungicide recommendations, including the appropriate chemical combination and the recommended timing for foliar spray.

During the field trials, fungicide application was therefore initiated only after DSS-based disease confirmation, whereas untreated control plots did not receive any fungicide intervention. The DSS did not recommend plot-wise spatial spraying within fields; instead, it functioned as a decision-support tool for disease confirmation and timely management intervention.

We acknowledge that the original wording may have implied a broader validation of precision agriculture capabilities. To avoid overstatement, we have revised the Methods and Results sections to clearly reflect that the field trials validate the effectiveness of DSS-guided advisory decisions (disease-triggered fungicide application) rather than the fungicide chemistry alone.

The manuscript has been revised accordingly to explicitly describe the DSS role and to align claims with the actual scope of validation.

Revisions made (highlighted in green in the revised manuscript)

• Materials and Methods – Section V: Avoidable Yield Loss in Maize through Advisory Measures for MDM Management:

Expanded to clearly describe how the DSS was used as a diagnostic and advisory trigger, specifying that fungicide application was initiated based on DSS-confirmed disease detection and stage-specific recommendations, rather than uniform or spatially targeted spraying.

• Results – Effectiveness of DSS-Guided Fungicide Application:

Revised to clarify that the observed disease reduction and yield gains reflect the impact of DSS-guided advisory-based intervention, not merely the standalone efficacy of the fungicide treatment.

Reviewer Comment 4 The authors should expand the Methods section (Section V.) to detail the actual protocol for the "DSS-guided fungicide applications". If the DSS only provided the chemical recommendation which was then applied uniformly, the claims about validating the Decision Support System need to be rephrased to accurately reflect what was tested.

Authors Response We thank the reviewer for this constructive suggestion and agree that the protocol for DSS-guided fungicide application required clearer description.

We clarify that the Decision Support System (DSS) did not function as an automated or spatial precision-spraying system. Instead, the DSS served as a diagnostic and advisory decision-support tool. The system analysed field-acquired maize leaf images using the trained VGG16 model to confirm the presence of maize downy mildew. Based on confirmed disease detection, the DSS provided advisory recommendations regarding the necessity and timing of fungicide application, including the recommended chemical combination.

During the field trials, fungicide application was therefore initiated only after DSS-based disease confirmation and applied uniformly to the designated treatment plots. Untreated control plots received no fungicide application. The objective of the field experiment was to validate the effectiveness of DSS-triggered advisory intervention (disease-confirmed and timely application), rather than to test variable-rate or plot-wise precision spraying.

To accurately reflect this scope, we have expanded Section V of the Materials and Methods to explicitly describe the DSS-guided protocol and have rephrased relevant claims in the Results and Discussion sections to avoid overstatement.

Revisions made (highlighted in green in the revised manuscript)

Materials and Methods – Section V:

Avoidable Yield Loss in Maize through Advisory Measures for MDM Management:

Expanded to explicitly detail the DSS-guided fungicide application protocol, clarifying that the DSS functioned as a diagnostic and advisory decision-support tool. The revised text specifies that fungicide application was initiated following DSS-confirmed disease detection and applied uniformly to treatment plots, while untreated control plots received no intervention.

Results and Discussion:

Rephrased to ensure that claims accurately reflect the validation of DSS-guided advisory-based intervention rather than automated or spatially explicit precision spraying.

Reviewer Comment 5 The manuscript states that thirteen ML and DL algorithms were evaluated, and the results section briefly discusses the performance of a few (Logistic Regression, KNN, DT, SVM, GBM, RF, RNN, LSTM, DBN, CNN, Siamese Networks, and DLTL). To enable a complete peer review and maximize the value of the comparative study, a comprehensive table should be included that lists all performance metrics (Accuracy, Precision, Recall, F1-Score, AUC-ROC) for all thirteen models.

Authors Response We thank the reviewer for this constructive suggestion. We agree that a consolidated presentation of performance metrics for all evaluated models improves clarity and facilitates complete peer review.

Accordingly, we have added a new comprehensive table summarizing the performance of all thirteen machine-learning and deep-learning models evaluated in this study. The table reports Accuracy, Precision, Recall, F1-score, and AUC–ROC for each model, thereby enabling a clear and direct comparison across algorithms.

As the comparative analysis of ML and DL models is presented first in the Results section, this table has been included as Table 1. The field-trial and yield-related results are now presented subsequently as Table 2. All in-text table citations have been updated accordingly to ensure consistency.

Reviewer Comment 6 The citation for regression metrics is referred to as Jadesha et al. [25a]. Please ensure this citation adheres to the journal's standard numerical or author-year format, as the [25a] suffix may indicate a non-standard reference convention.

Authors Response We thank the reviewer for pointing out this citation formatting issue. We agree that the use of the suffix [25a] does not conform to the journal’s standard numerical citation style.

The non-standard citation has been corrected by removing the suffix and replacing it with the appropriate numerical references ([21,22]), which are already included in the reference list. The manuscript has been revised to ensure consistent and standard citation formatting throughout.

Revisions made (highlighted in green in the revised manuscript)

• Materials and Methods – Regression Metrics Analysis of VGG16:

Corrected the in-text citation by removing the non-standard [25a] suffix and replacing it with standard numerical citations [21,22], in accordance with the journal’s referencing guidelines.

• References section:

Verified that all references follow a consistent numerical citation format without alphabetical suffixes.

Reviewer 2 Comments and Authors Response

We sincerely thank Reviewer 2 for the detailed and constructive assessment of our manuscript, which has greatly contributed to improving its clarity, scope alignment, and technical rigor. All comments and suggestions from Reviewer 2 have been carefully addressed and are explained in detail below. The corresponding changes have been highlighted in Blue in the revised manuscript. For ease of reference, revisions made in response to Reviewer 1 are highlighted in Green.

Reviewer #2: The manuscript presents a comprehensive, technically sound, and practically validated AI-based DSS for maize downy mildew. The integration of deep learning, explainability, and field-based validation significantly strengthens its contribution. Minor clarifications and formatting improvements are required before acceptance.

Clarifications Required

Reviewer Comment 1 1. Novelty Clarification

The manuscript presents a comprehensive AI-based framework for maize downy mildew detection and management. However, the novelty of the study can be stated more explicitly.

Please add one clear sentence in the Introduction highlighting how this work differs from existing studies, particularly in terms of integration of deep learning, explainability, and field-validated decision support.

Authors Response We thank the reviewer for this valuable suggestion. We agree that the novelty of the study can be articulated more explicitly. Accordingly, we have added a clear sentence in the Introduction highlighting how the present work differs from existing studies by integrating deep-learning–based disease detection with explainable AI analyses and validating a web-enabled decision support system through multi-season field trials. This addition clarifies the unique contribution of the study beyond standalone model development.

Revisions made (highlighted in blue in the revised manuscript)

• Introduction:

Added a single sentence explicitly stating the novelty of the proposed framework, emphasizing the integration of deep learning, explainable AI, and fi

---

## [Decision Letter · Decision Letter 1]

8 Feb 2026

Artificial Intelligence–Driven Detection and Decision Support System for Precision Management of Maize Downy Mildew

PONE-D-25-55507R1

Dear Dr. D,

We’re pleased to inform you that your manuscript has been judged scientifically suitable for publication and will be formally accepted for publication once it meets all outstanding technical requirements.

Kind regards,

C Anilkumar, Ph.D.

Academic Editor

PLOS One

Additional Editor Comments (optional):

Dear Authors,

Thanks for revising the manuscript. All reviewer concerns have now been addressed satisfactorily. I recommend acceptance of the manuscript for publication.

Thank you

Reviewers' comments:

Reviewer's Responses to Questions

**Comments to the Author**

Reviewer #1: All comments have been addressed

2. Is the manuscript technically sound, and do the data support the conclusions?

Reviewer #1: Yes

3. Has the statistical analysis been performed appropriately and rigorously?

Reviewer #1: Yes

4. Have the authors made all data underlying the findings in their manuscript fully available?

Reviewer #1: Yes

5. Is the manuscript presented in an intelligible fashion and written in standard English?

Reviewer #1: Yes

Reviewer #1: My comments have been addressed. Given the relevance of the topic, the article can be accepted for publication.

**Do you want your identity to be public for this peer review?** For information about this choice, including consent withdrawal, please see our Privacy Policy

Reviewer #1: **Yes:** Andronicus A. Akinyelu

---

## [Editor Report · Acceptance letter]

PONE-D-25-55507R1

PLOS One

Dear Dr. D,

I'm pleased to inform you that your manuscript has been deemed suitable for publication in PLOS One. Congratulations! Your manuscript is now being handed over to our production team.

Kind regards,

on behalf of

Dr. C Anilkumar

Academic Editor

PLOS One